# Quantification of Volatile Aldehydes Deriving from In Vitro Lipid Peroxidation in the Breath of Ventilated Patients

**DOI:** 10.3390/molecules26113089

**Published:** 2021-05-21

**Authors:** Lukas Martin Müller-Wirtz, Daniel Kiefer, Sven Ruffing, Timo Brausch, Tobias Hüppe, Daniel I. Sessler, Thomas Volk, Tobias Fink, Sascha Kreuer, Felix Maurer

**Affiliations:** 1CBR—Center of Breath Research, Department of Anaesthesiology, Intensive Care and Pain Therapy, Saarland University Medical Center, Homburg, 66421 Saarland, Germany; daniel.kiefer@uks.eu (D.K.); sven.ruffing@uks.eu (S.R.); timo.brausch@uks.eu (T.B.); tobias.hueppe@uks.eu (T.H.); thomas.volk@uks.eu (T.V.); tobias.fink@uks.eu (T.F.); sascha.kreuer@uks.eu (S.K.); felix.maurer@uks.eu (F.M.); 2Outcomes Research Consortium, Cleveland, OH 44195, USA; ds@or.org; 3Department of Outcomes Research, Anesthesiology Institute, Cleveland Clinic, Cleveland, OH 44195, USA

**Keywords:** anesthesia, breath analysis, mechanical ventilation, lipid peroxidation, biomarker, volatile aldehydes, pentanal, MCC–IMS, ventilator-induced lung injury, volatile organic compounds

## Abstract

Exhaled aliphatic aldehydes were proposed as non-invasive biomarkers to detect increased lipid peroxidation in various diseases. As a prelude to clinical application of the multicapillary column–ion mobility spectrometry for the evaluation of aldehyde exhalation, we, therefore: (1) identified the most abundant volatile aliphatic aldehydes originating from in vitro oxidation of various polyunsaturated fatty acids; (2) evaluated emittance of aldehydes from plastic parts of the breathing circuit; (3) conducted a pilot study for in vivo quantification of exhaled aldehydes in mechanically ventilated patients. Pentanal, hexanal, heptanal, and nonanal were quantifiable in the headspace of oxidizing polyunsaturated fatty acids, with pentanal and hexanal predominating. Plastic parts of the breathing circuit emitted hexanal, octanal, nonanal, and decanal, whereby nonanal and decanal were ubiquitous and pentanal or heptanal not being detected. Only pentanal was quantifiable in breath of mechanically ventilated surgical patients with a mean exhaled concentration of 13 ± 5 ppb. An explorative analysis suggested that pentanal exhalation is associated with mechanical power—a measure for the invasiveness of mechanical ventilation. In conclusion, exhaled pentanal is a promising non-invasive biomarker for lipid peroxidation inducing pathologies, and should be evaluated in future clinical studies, particularly for detection of lung injury.

## 1. Introduction

Lipid peroxidation products are established markers of oxidative stress [1], and are potential non-invasive biomarkers for detection of various diseases. For example, increased aldehyde exhalation has been reported in patients suffering from pulmonary diseases such as lung cancer [2,3,4,5], chronic obstructive pulmonary disease [6], and COVID-19 [7,8]. Additionally, volatile aldehydes considerably increase in the blood of patients suffering from acute respiratory distress syndrome [9,10]. Recent animal experiments suggest that the volatile aldehyde pentanal may be a biomarker for ventilator-induced lung injury [11]. There is thus increasing evidence that monitoring of aldehyde exhalation may help detect diseases and acute injuries of the lung.

Analyses of liquid aliphatic aldehydes indicate that they originate from lipid peroxidation [12,13]. However, gaseous concentrations also depend on vapor pressure, which progressively decreases with longer aldehyde chain lengths. Most previous investigations that measured gaseous concentrations of volatile aliphatic aldehydes focused on single aldehydes to quantify lipid peroxidation in vitro [14,15]. To our knowledge, only a single study performed comparative measurements of various volatile aldehydes deriving from oxidizing synthetic lipid membranes [16]. Consequently, the relative contributions of various isolated polyunsaturated fatty acids to gaseous aldehyde generation remains unclear.

Monitoring aldehyde exhalation is particularly interesting in ventilated patients, as they often have baseline pulmonary diseases and are susceptible to ventilator-induced lung injury, which might be identified by aldehyde exhalation [11]. A potential complication, though, is that most materials in breathing circuits of modern anesthesia machines and airway devices are made from plastic which can emit volatile aldehydes [17], thus potentially interfering with breath analysis in ventilated patients.

Multicapillary column–ion mobility spectrometry (MCC–IMS) has been used to examine exhaled volatile organic compounds [18,19,20] and monitor exhaled propofol [21,22,23] in ventilated patients. However, volatile aldehydes are typically exhaled at concentrations of a few parts per billion [24,25], which raises the question of whether the MCC–IMS technique with our corresponding sampling setup is sensitive enough to quantify exhaled aldehydes. Moreover, cross-contaminations from ambient air or the ventilator may even exclude in vivo quantification. A pilot study is therefore needed to assess the clinical suitability of the MCC–IMS technique to monitor aldehyde exhalation.

Our primary aim was to assess clinical use of MCC–IMS for bedside online measurements of exhaled aliphatic aldehydes as a measure of lipid peroxidation. Secondarily, we aimed to identify the most promising aliphatic aldehydes for monitoring lipid peroxidation in mechanically ventilated patients under in vitro and in vivo conditions. We therefore: (1) identified the predominant volatile aliphatic aldehydes originating from in vitro peroxidation of various polyunsaturated fatty acids; (2) evaluated emittance of volatile aldehydes from parts of the breathing circuit; (3) conducted a pilot study using MCC–IMS for in vivo quantification of exhaled aldehydes in mechanically ventilated patients.

## 2. Results

### 2.1. Calibration

All calibrations showed a good linear fit (R^2^: 0.97 to 0.99; Appendix A). Limits of detection and quantification were 0.008 and 0.011 Volt.

### 2.2. Volatile Aldehydes Originating from In Vitro Lipid Peroxidation

Pentanal, hexanal, heptanal, and nonanal were detected in the headspace of an animal-sourced mixture of oxidizing polyunsaturated fatty acids (PUFA-Mix, Figure 1). Pentanal and hexanal emerged from all polyunsaturated fatty acids, with pentanal and hexanal predominating. Nonanal emerged from the mixture of polyunsaturated fatty acids and heptanal from arachidonic acid (Figure 1). Unquantifiable traces of octanal were identified in all probes. Decanal was not detected.

### 2.3. Volatile Aldehydes Emitted By Plastic Parts of the Breathing Circuit

Parts of the breathing circuit emitted hexanal, octanal, nonanal, and decanal with nonanal and decanal originating from all assessed parts (Table 1). Pentanal and heptanal were not detected.

### 2.4. Volatile Aldehydes in the Breath of Ventilated Patients

All patients screened for eligibility underwent pancreaticoduodenectomies, as this operation is scheduled to last at least 4 h at our medical center. Twelve surgical adult patients undergoing elective pancreaticoduodenectomy were assessed. No patient was excluded.

Two third of the patients had a malignant tumor and/or arterial hypertension, and half of the patients suffered from diabetes mellitus. Patients were ventilated on average for about six hours (Table 2).

Only exhaled pentanal was quantifiable in breath. Unquantifiable traces of nonanal were detected and other aldehydes were not detected at all. Ventilators were contaminated with a mean pentanal concentration of 1.2 ± 1.1 ppb. Exhaled pentanal concentrations over time are presented in Figure 2. In three patients, the exhaled pentanal concentration did not substantially exceed ventilator contaminations who were therefore excluded from further statistical analyses. None of the patients with limited pentanal exhalation had malignant tumors. In contrast, all but one of the remaining nine patients had malignancy. Excluding the three patients with limited pentanal exhalation, the overall mean exhaled pentanal concentration was 13 ± 5 ppb.

An exploratory analysis revealed a significant association of exhaled pentanal with tidal volume, minute volume, and mechanical power, but not with inspiratory pressure (Figure 3, Table 3).

## 3. Discussion

This study presents the preparatory analytical work and a clinical pilot study for online monitoring of aldehyde exhalation to quantify lipid peroxidation in ventilated surgical patients by means of MCC–IMS. As an overall finding, pentanal represents the most promising exhaled volatile aliphatic aldehyde. Pentanal is predominantly a product of lipid peroxidation rather than evaporating from parts of the breathing circuit and was the only quantifiable volatile aldehyde in the breath of ventilated surgical patients with our measurement setup.

Oxidation of mixed polyunsaturated fatty acids confirmed that the aliphatic aldehydes pentanal, hexanal, heptanal, and nonanal are gaseous products of lipid peroxidation. Only traces of octanal were detected and no decanal was detected, probably because they were generated in limited quantities and their vapor pressures are low [26]. In contrast, pentanal and hexanal were ubiquitous. Pentanal dominated early phases of in vitro lipid peroxidation, whereas hexanal increased over time and approached or even exceeded pentanal concentrations. Our findings are consistent with a previous study that assessed volatile aldehydes emerging from oxidizing phospholipid membranes and similarly showed that pentanal dominated early in the process of lipid peroxidation, whereas hexanal dominated later [16]. Results from an analysis of fluid aliphatic aldehydes, originating from several polyunsaturated fatty acids oxidized by air, confirm that pentanal and hexanal are the predominant products of lipid peroxidation, with hexanal concentrations being twice those of pentanal after 48 h of peroxidation [12].

Pentanal thus dominates early and hexanal dominates later phases of lipid peroxidation, which can be explained by their chemical properties. Specifically, pentanal has twice the vapor pressure than hexanal (pentanal: 26 mmHg at 20 °C and hexanal: 11.3 mmHg at 25 °C), and therefore evaporates more quickly [27,28]. However, pentanal is more reactive then hexanal. Consequently, autoxidation of pentanal may, over time, exceed its generation rate. Hexanal may therefore be the primary and more stable product of lipid peroxidation, but pentanal may be a better biomarker by virtue of responding quickly to oxidative stress.

Plastic components of the breathing circuit emitted hexanal, octanal, nonanal, and decanal, which is consistent with previous analyses showing that all are emitted by polypropylene and polyethylene [17,29]—the two most commonly used materials for plastic components. The largest source of volatile aldehydes was the endotracheal tube, which is made from polyvinylchloride. In addition to octanal and decanal, the endotracheal tube emitted considerable amounts of nonanal, which is consistent with a previous analysis of volatile organic compound profiles emitted by polyvinylchloride materials [30]. Under the influence of heat and moisture from the body and breathing gases, release of these volatile organic compounds from plastic components may be unpredictable. Thus, even when corrected for baseline contamination, measurements of hexanal, octanal, nonanal, and decanal in the breath of ventilated patients might be compromised by non-organic sources. In contrast, pentanal and heptanal were not emitted by plastic breathing circuit components and are thus presumably better biomarkers for lipid peroxidation in ventilated patients.

We finally evaluated aldehyde exhalation in twelve surgical patients during prolonged mechanical ventilation. Aside from traces of nonanal, pentanal was the only volatile aldehyde we detected. Ventilators were contaminated with low amounts of pentanal, possibly representing residuals from previously ventilated patients since the breathing circuit was apparently not the source. The overall exhaled pentanal concentrations in ventilated patients were in the low parts-per-billion range, consistent with previous reports from spontaneously breathing healthy volunteers [24,25]. However, we measured slightly higher exhaled pentanal concentrations in ventilated patients, possibly consequent to intubation, which increases the sampled proportion of alveolar air. Another reason could be that mechanical ventilation induces pulmonary lipid peroxidation and therefore increase pentanal exhalation, as previously shown in animals [11,31].

In three patients, the exhaled pentanal concentrations were roughly at the concentration of ventilator contamination, whereas exhaled concentrations in the others averaged 13 ppb. Interestingly, none of the patients with low concentrations had cancer, whereas eight of the nine others did. While possibly spurious, the results are consistent with previous reports that cancer promotes exhalation of pentanal [2,3,4,32,33].

Although we did not actively vary ventilation parameters, our explorative analysis revealed a significant linear relationship between exhaled pentanal and mechanical power—a clinical measure for the invasiveness of mechanical ventilation [34,35]. The higher the mechanical power dissipated to the lungs, the higher was the exhaled pentanal concentration, which is consistent with our previous findings in 150 ventilated rats [11]. We therefore previously proposed that exhaled pentanal results from stretched lung tissue, which exposes lipids in cell membranes to oxidation. Intuitively, higher minute volumes may dilute exhaled pentanal. Instead, higher minute volumes were associated with increased pentanal exhalation, further supporting the hypothesis that mechanical ventilation induces pulmonary lipid peroxidation measurable by exhaled pentanal—a potential biomarker for ventilator-induced lung injury. More experimental and clinical studies are needed to identify various causes of pentanal exhalation, which is critical to its potential use as a biomarker.

A limitation of our study is that a mixture of exhaled and inspired gases can lead to cross-contaminations or diluted concentrations. An integration of carbon dioxide or flow triggered sampling could help sample isolated exhaled gas and thus increase the proportion of alveolar gas in breath samples [36]. Furthermore, activated charcoal filters, originally designed to eliminate residual volatile anesthetics emitted from anesthesia workstations, are now available [37]. Using an activated charcoal filter between the anesthesia machine and the inspiratory limb of the rebreathing circuit would presumably eliminate contamination from within the machine. We also note that our study population is small. While sufficient to confirm applicability of our measurement setup to patients, larger studies will be necessary to confirm the association of pentanal exhalation and mechanical power.

In summary, pentanal and hexanal are the predominant volatile aldehydes deriving from lipid peroxidation under in vitro conditions, and therefore represent promising breath biomarkers for oxidative stress. Emission of volatile aldehydes from plastic parts of the breathing circuit may bias breath analysis for hexanal, octanal, nonanal, and decanal but not for pentanal. Future studies should quantify exhaled pentanal in mechanically ventilated patients with various pathologies and assess its potential as a biomarker for ventilator-induced lung injury.

## 4. Materials and Methods

### 4.1. Calibration

The detailed experimental setup and procedure of the calibration is presented in the supplement (Appendix A). In short, hexane-diluted aldehyde standards were pipetted into a closed flask made from inert perfluoroalkoxy alkane. Evaporation was accelerated by an electrically driven fan inside the flask and the resulting gaseous mixture was sampled by the MCC–IMS (B&S Analytik, Dortmund, Germany). The composition of the liquid hexane-diluted aldehyde standards needed to generate specific gaseous concentrations inside the flask were calculated according to the ideal gas law (Appendix A).

### 4.2. Volatile Aldehydes Originating from In Vitro Lipid Peroxidation

We used the same technical setup as for calibration (Appendix A). A total of 30 µL of an animal-sourced mixture of polyunsaturated fatty acids and three isolated polyunsaturated fatty acids—linoleic, linolenic, and arachidonic acid (analytical standard, Merck, Darmstadt, Germany)—were oxidized in a cleaned flask under 100 mL/min flow of highly purified synthetic air (oxygen content: 21%; Alphagaz 1, Air Liquide, Paris, France) and constant fanning. Headspace gas was sampled at 10-min intervals by the MCC–IMS. Signal intensities between the limit of detection and quantification were considered as unquantifiable traces.

### 4.3. Volatile Aldehydes Emitted by Plastic Parts of the Breathing Circuit

An endotracheal tube (Ruesch^®^, Teleflex, Kernen, Germany) and a humidity and moisture exchanging filter (Gibeck Humid-Vent^®^; Teleflex, Kernen, Germany) were placed in the cleaned flask used for calibration and lipid peroxidation. The flask was sealed and flushed with purified nitrogen for 5 min and subsequently with highly purified synthetic air for 1 min. Breathing tubes and bag (Anesthesia set VentStar^®^, disposable, basic, 2 L, 1.8 m/1.5 m, latex-free, Draeger, Lübeck, Germany) and a test lung (Draeger SelfTestLung^TM^) were flushed from the inside using a similar procedure. Headspace gas was sampled from materials placed in the flask and from the inside of the breathing tube, breathing bag, and test lung by the MCC–IMS at 5-min intervals for at least 20 min. All measurements were performed in a room maintained at 20 °C with an air purification system (CamCleaner City M, Camfil, Reinfeld, Germany).

### 4.4. Volatile Aldehydes in the Breath of Ventilated Patients

#### 4.4.1. Ethics

Clinical investigations were approved by the local ethics commission (No. 81/19, Ärztekammer des Saarlandes, Saarbücken, Germany), and written informed consent was obtained.

#### 4.4.2. Inclusion and Exclusion Criteria

We included patients aged >18 years, American Society of Anesthesiologists (ASA) physical status <4, body mass index (BMI) ≤35 kg/m^2^ and scheduled for general surgery expected to last about 4 h. Patients with mental disorders, drug abuse, human immunodeficiency virus or hepatitis infection, isolation requirement, pregnancy, or any contraindication for total intravenous anesthesia were excluded.

#### 4.4.3. Measurements

Twelve patients were anesthetized with propofol and remifentanil, and pressure-controlled ventilation was maintained at lung protective settings (tidal volume: 6–8 mL/kg, maximum inspiratory pressure: 30 mbar, ventilator: Primus, Draeger, Luebeck, Germany). The ventilator’s fresh gas flow was set to 1 L/min throughout the case. The choice of the inspiratory oxygen concentration was left to the attending anesthetist. A new set of breathing tubes and bag was used for each case.

The MCC–IMS was connected to a t-piece at the tracheal tube by 1.8-m-long perfluoroalkoxy alkane tubing. Exhaled gas was sampled at 5-min intervals. Ventilator contamination was assessed during ventilation of a test lung prior to patient assessment. The mean of the three final concentrations was defined as baseline contamination, which was subtracted from measured concentrations for the relevant patient. Ventilation variables were electronically captured from the ventilators by specific software programmed by Bertram Bödecker.

Mechanical power was calculated with the following formula: mechanical power (J·min^−1^) = 0.098 × respiratory rate (breaths·min^−1^) × tidal volume (mL) × (positive end-expiratory pressure (cmH_2_O) + driving pressure (cmH_2_O)) [38].

### 4.5. Statistics

VoCan 3.7 (B&S Analytik, Dortmund, Germany) was used for MCC–IMS device control and Visual Now 3.7 for spectrum analysis (B&S Analytik). Statistical analyses were carried out with R 4.0.2 (R Core Team, 2020) using the packages geepack (Højsgaard, Halekoh, and Yan, 2006) and broom (v0.7.5; Robinson, Hayes, and Couch, 2021). Figures were created with SPSS 26 (IBM, Armonk, NY, USA). Total intensities for a compound in volts were calculated by summing the intensity of the monomer and twice the intensity of the dimer. Calibration formulas were estimated by linear regression. If the relative standard deviation of measurements of any standard exceeded 20%, Dixon’s test was used to exclude outliers [39]. Limits of detection (LOD) and quantification (LOQ) were calculated from background noise intensities as follows: LOD = mean + 3 × SD; LOQ = mean + 10 × SD.

Normality of data distribution was confirmed by visual assessment of histograms and quantile–quantile plots. Aldehyde generation by oxidizing polyunsaturated fatty acids was assessed once; therefore, raw data is presented. Repeatedly measured volatile aldehyde concentrations evaporated by parts of the breathing circuit are presented as means ± SD. Exhaled pentanal concentrations are presented graphically as means ± SD and additionally as a mean across all patients and time with the corresponding standard deviation. The relationship of exhaled pentanal with ventilation parameters and mechanical power was assessed by linear generalized estimating equations regression. The marginal R^2^ was calculated according to Zheng’s method [40]. Due to the explorative character of the clinical investigations, there was no a priori sample size estimation.

## Figures and Tables

**Figure 1 molecules-26-03089-f001:**
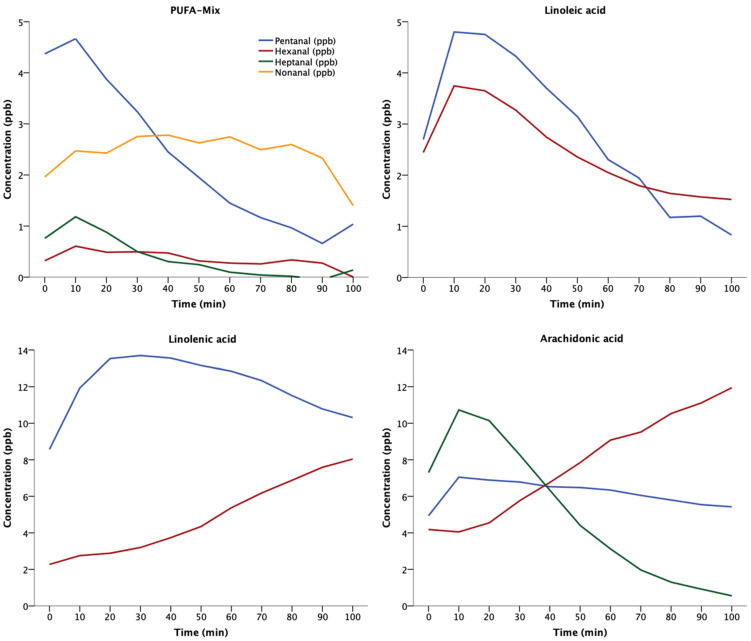
Volatile aldehydes produced by oxidation of polyunsaturated fatty acids. A total of 30 µL of isolated or mixed polyunsaturated fatty acids were injected into a perfluoroalkoxy alkane flask and oxidized under a constant flow of 100 mL/min synthetic air (21% O2). Headspace samples were analyzed by means of MCC–IMS. Measurement series were performed once; therefore, raw data are presented. PUFA-Mix is the animal-sourced mixture of polyunsaturated fatty acids.

**Figure 2 molecules-26-03089-f002:**
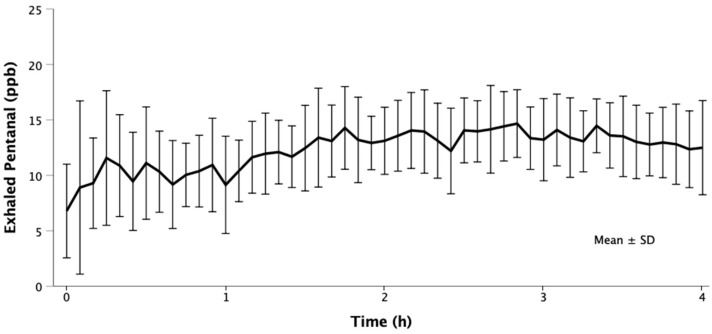
Pentanal in the breath of surgical patients during mechanical ventilation. Twelve surgical patients undergoing prolonged mechanical ventilation (≥4 h) were assessed. In three patients, the exhaled pentanal concentration did not substantially exceed ventilator contaminations who were therefore excluded from graphical presentation. The presented concentrations are corrected for baseline ventilator contaminations.

**Figure 3 molecules-26-03089-f003:**
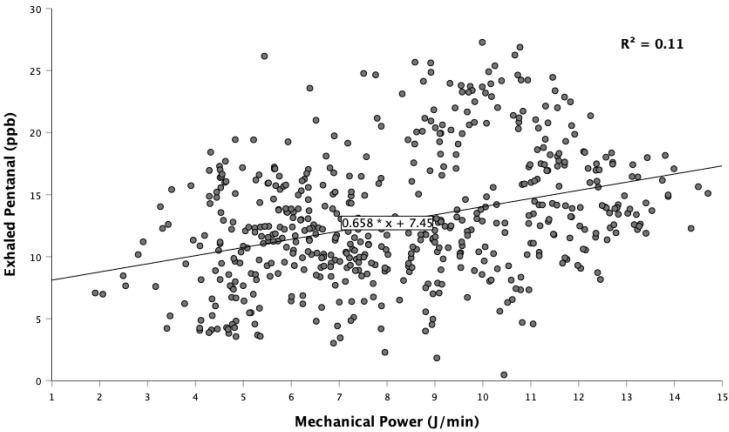
Exhaled pentanal versus mechanical power. A linear generalized estimating equations regression model and the marginal R^2^ were calculated to assess the relationship of exhaled pentanal with mechanical power. Subjects included/total: *n* = 9/12, data pairs: *n* = 547. Data from three patients with limited pentanal exhalation were excluded.

**Table 1 molecules-26-03089-t001:** Evaporation of volatile aldehydes by parts of the breathing circuit.

Material	Detected Aldehydes	Concentration (ppb)
Endotracheal tube	Octanal	7.0 ± 1.4
	Nonanal	12.5 ± 0.7
	Decanal	2.5 ± 0.4
Humidity and moisture exchanging filter	Nonanal	0.1 ± 0.4
	Decanal	1.7 ± 0.1
Breathing bag	Hexanal	0.5 ± 0.1
	Nonanal	2.0 ± 0.3
	Decanal	0.7 ± 0.1
Breathing tubes	Hexanal	0.2 ± 0.1
	Nonanal	1.4 ± 0.2
	Decanal	0.6 ± 0.2
Test lung	Nonanal	unquantifiable traces
	Decanal	0.8 ± 0.2

Data are presented as means ± SD.

**Table 2 molecules-26-03089-t002:** Patient characteristics and ventilation parameters.

**Patient Characteristics**	
Patients included/screened for eligibility	12/12
Age (years)	67 ± 11
Sex (male/female)	8 (67)/4 (33)
Height (cm)	170 ± 8
Weight (kg)	69 ± 13
ASA physical status (I/II/III)	0/6 (50)/6 (50)
Malignant tumor	8 (67)
Arterial hypertension	8 (67)
Diabetes mellitus	6 (50)
Mechanical ventilation time (min)	344 ± 102
**Ventilation Parameters**	
Tidal volume (mL)	452 ± 82
Respiratory rate (breaths·min^−1^)	12 ± 1
Minute volume (L·min^−1^)	5.4 ± 1.1
Inspiratory pressure (mbar)	15.3 ± 2.1
Positive end expiratory pressure (mbar)	5.1 ± 0.5
Mechanical power (J·min^−1^)	8.3 ± 2.6

Data are presented either as means ± SD, or as frequencies (%). Ventilation parameters repeatedly measured over time were summarized with single means.

**Table 3 molecules-26-03089-t003:** Association of exhaled pentanal with ventilation parameters.

Parameter	Regression Coefficient	95% Confidence Interval	R²	*p*
Tidal volume (mL)	0.01	0.003–0.018	0.02	0.004
Minute volume (L·min^−1^)	2.0	0.6–3.3	0.05	0.004
Inspiratory pressure (mbar)	0.2	−0.3–0.6	0.04	0.463
Mechanical power (J·min^−1^)	0.7	0.3–1.1	0.11	0.001

Univariable linear generalized estimating equations regression models were calculated to assess the association of exhaled pentanal (dependent variable) with ventilation parameters and mechanical power (independent variable). Subjects: *n* = 9, data pairs: *n* = 547. Data from 3 patients with limited pentanal exhalation were excluded.

## Data Availability

Data is contained within the article or Appendix A. The data on intraoperative exhaled pentanal measurements and ventilation parameters are available in Appendix A.

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
