# Peer review of "Quantification of Volatile Aldehydes Deriving from In Vitro Lipid Peroxidation in the Breath of Ventilated Patients"

_molecules, 2021, doi:10.3390/molecules26113089_

Round 1

Reviewer 1 Report

This manuscript presents the analysis of volatile aldehydes (pentanal, hexanal, heptanal, and nonanal) in breath of ventilated patients. These aldehydes were also measured in the emissions of breathing circuit parts. The authors found that exhaled pentanal can be a potential biomarker for lung injury assessment.  It is an important pilot study. It highlights the significance of future research on analysis of breath aldehydes.

The manuscript is well organized and well written. I have a few major and minor comments:

Major comments

  1. Even though this is a pilot study, a very small number of patients was tested. The authors should highlight this limitation inthe text. Overall, the paragraph on “limitations of the present study” is missing and should be added.
  2. I am wondering, why low molecular weight aldehydes were not analyzed (formaldehyde, acetaldehyde, propanal, butanal, etc.)? It would be beneficial for the current study, since low molecular weight breath aldehydes can also serve as biomarkers of oxidative stress.
  3. Error bars should be added to Figure 1 and calculation of SD should be explained.

Some minor comments

Figure 2. Ticks are missing on Y-axis

  1. 3, PUFA abbreviation has to be explained and used in the text, otherwise it should be removed from the manuscript
  2. 8, line 263. HIV abbreviation has to be explained

Fig.2 Fonts are too small

In summary, I recommend this manuscript for publication after major revisions

Author Response

Reviewer #1)

This manuscript presents the analysis of volatile aldehydes (pentanal, hexanal, heptanal, and nonanal) in breath of ventilated patients. These aldehydes were also measured in the emissions of breathing circuit parts. The authors found that exhaled pentanal can be a potential biomarker for lung injury assessment. It is an important pilot study. It highlights the significance of future research on analysis of breath aldehydes.

The manuscript is well organized and well written. I have a few major and minor comments:

Major comments

  1. Even though this is a pilot study, a very small number of patients was tested. The authors should highlight this limitation in the text. Overall, the paragraph on “limitations of the present study” is missing and should be added.

Authors: We added the limitations of the current study at the end of the discussion section and highlighted the small number of patients included in the study:

“A limitation of our study is that a mixture of exhaled and inspired gases can lead to cross-contaminations or diluted concentrations. An integration of carbon dioxide or flow triggered sampling could help sample isolated exhaled gas and thus increase the proportion of alveolar gas in breath samples [35]. Furthermore, activated charcoal filters, originally designed to eliminate residual volatile anesthetics emitted from anesthesia workstations, are now available [36]. Using an activated charcoal filter between the anesthesia machine and the inspiratory limb of the rebreathing circuit would presumably eliminate contamination from within the machine. We also note that our study population is small. While sufficient to confirm applicability of our measurement setup to patients, larger studies will be necessary to confirm the association of pentanal exhalation and mechanical power.”

  1. I am wondering, why low molecular weight aldehydes were not analyzed (formaldehyde, acetaldehyde, propanal, butanal, etc.)? It would be beneficial for the current study since low molecular weight breath aldehydes can also serve as biomarkers of oxidative stress.

Authors: We agree that aldehydes with shorter chain length are also interesting to investigate in the context of oxidative stress. However, with our MCC-IMS devices, such small molecules cannot be reliably detected because they appear close to the reactant ion peak and often other peaks overlie them.

  1. Error bars should be added to Figure 1 and calculation of SD should be explained.

Authors: Volatile aldehydes produced by oxidation of polyunsaturated fatty acids were measured only once so raw data are presented. We apologize for the confusion and clarified this in the legend of Figure 1: “Measurement series were performed once; therefore, raw data are presented.”

Some minor comments

Figure 2. Ticks are missing on Y-axis

Authors: Ticks were added in the revised version.

  1. 3, PUFA abbreviation has to be explained and used in the text, otherwise it should be removed from the manuscript

Authors: We now mention “PUFA” in the results text: “Pentanal, hexanal, heptanal, and nonanal were detected in the headspace of an animal-sourced mixture of oxidizing polyunsaturated fatty acids (PUFA-Mix, Figure 1).” Its meaning is also specified in the legend of Figure 1: “PUFA-Mix, mixture of polyunsaturated fatty acids.”

  1. 8, line 263. HIV abbreviation has to be explained

Authors: HIV is now spelled out in the text.

Fig.2 Fonts are too small

Authors: Figures 2 and 3 were previously decreased in size to fit on one page. We now enlarged both figures to their original size.

In summary, I recommend this manuscript for publication after major revisions.

Reviewer 2 Report

The idea behind the study is very interesting. 

However i must say that the manuscript is somewhat unclear  about different points and should be (partly) modified to become more understandable.

In detail:

  • A subchapter about patients selection, inclusion/exclusion criteria, numbero of included patients and (if any) excluded patients and the reason. Also, Table 2 should be modified accordingly.
  • A graphical representation of the study design would help readers to better understand the study. 
  • The aim of the study is missing. Please modify introduction and clearly state what's the study purpose. 
  • Did authors estimate sample size? Please specify it.
  • I found some english errors throughout the manuscript. Please have a re-check

Author Response

The idea behind the study is very interesting. 

However, I must say that the manuscript is somewhat unclear about different points and should be (partly) modified to become more understandable.

In detail:

  • A subchapter about patients’ selection, inclusion/exclusion criteria, numbers of included patients and (if any) excluded patients and the reason. Also, Table 2 should be modified accordingly.

Authors: We separated the chapter “4.4) Volatile aldehydes in the breath of ventilated patients” into several subchapters including a subchapter about “4.4.2) Inclusion and exclusion criteria”. Table 2 now states the number of patients included and screened for eligibility.

  • A graphical representation of the study design would help readers to better understand the study. 

Authors: We have created a graphical abstract to enable readers to understand the study design and our major finding with one view. Please find it attached.

  • The aim of the study is missing. Please modify introduction and clearly state what's the study purpose. 

Authors: We modified the last paragraph of the introduction to clearly state the aim of the study:

“Our primary aim was to assess clinical use of MCC-IMS for bedside online measurements of exhaled aliphatic aldehydes as a measure of lipid peroxidation. Secondarily, we aimed to identify the most promising aliphatic aldehydes for monitoring lipid peroxidation in mechanically ventilated patients under in vitro and in vivo conditions. We therefore: 1) identified the predominant volatile aliphatic aldehydes originating from in vitro peroxidation of various polyunsaturated fatty acids; 2) evaluated emittance of volatile aldehydes from parts of the breathing circuit; and, 3) conducted a pilot study using MCC-IMS for in vivo quantification of exhaled aldehydes in mechanically ventilated patients.”

  • Did authors estimate sample size? Please specify it.

Authors: We did not perform a sample size estimation for this pilot study. This is now specified at the end of the statistics chapter: “Due to the explorative character of the clinical investigations, there was no a priori sample size estimation.”

  • I found some English errors throughout the manuscript. Please have a re-check

Authors: Thank you for your thorough review of our manuscript. We reviewed the manuscript, and it should no longer contain English errors.

Round 2

Reviewer 1 Report

My questions and comments were well addressed by the author and I recommend this manuscript for publication

Reviewer 2 Report

for me now i'ts ok for publication